# miR-410-3p is induced by vemurafenib via ER stress and contributes to resistance to BRAF inhibitor in melanoma

**Tomasz M. Grzywa**[1,2], **Klaudia Klicka**[1], **Wiktor Paskal**[1]*, **Julia Dudkiewicz**[1], **Jarosław Wejman**[3], **Michał Pyzlak**[3], **Paweł K. Włodarski**[1]

**1** Department of Methodology, Laboratory of Centre for Preclinical Research, Medical University of Warsaw, Warsaw, Poland, **2** Department of Immunology, Medical University of Warsaw, Warsaw, Poland, **3** Department of Pathology, Medical Center of Postgraduate Education, Warsaw, Poland

* wiktor.paskal@wum.edu.pl

**Data Availability Statement:** All relevant data are within the paper and its Supporting Information files.

## Abstract

Despite significant development of melanoma therapies, death rates remain high. Micro-RNAs, controlling posttranscriptionally gene expression, play role in development of resistance to BRAF inhibitors. The aim of the study was to assess the role of miR-410-3p in response to vemurafenib–BRAF inhibitor. FFPE tissue samples of 12 primary nodular melanomas were analyzed. With the use of Laser Capture Microdissection, parts of tumor, transient tissue, and adjacent healthy tissue were separated. In vitro experiments were conducted on human melanoma cell lines A375, G361, and SK-MEL1. IC50s of vemurafenib were determined using MTT method. Cells were transfected with miR-410-3p mimic, anti-miR-410-3p and their non-targeting controls. ER stress was induced by thapsigargin. Expression of isolated RNA was determined using qRT-PCR. We have found miR-410-3p is downregulated in melanoma tissues. Its expression is induced by vemurafenib in melanoma cells. Upregulation of miR-410-3p level increased melanoma cells resistance to vemurafenib, while its inhibition led to the decrease of resistance. Induction of ER stress increased the level of miR-410-3p. miR-410-3p upregulated the expression of AXL in vitro and correlated with markers of invasive phenotype in starBase. The study shows a novel mechanism of melanoma resistance. miR-410-3p is induced by vemurafenib in melanoma cells via ER stress. It drives switching to the invasive phenotype that leads to the response and resistance to BRAF inhibition.

## Introduction

Melanoma is a skin cancer that derives from melanocytes. It is the deadliest type of skin cancer with the incidence rate of 19.7 per 100,000 and the age-adjusted death rate 2.7 per 100,000. Despite of awareness of well-known risk factors such as sunburn, long sun exposure, and indoor tanning, incidence rates continue to increase [1].

In recent years a significant improvement of melanoma therapies based on targeted therapy and immune therapy is observed, but still, an estimated median survival for patients with

**Funding:** This work was funded by a grant from the Medical University of Warsaw 1M15/NM5/18 (TMG) (www.wum.edu.pl) and from the Polish Ministry of Science and Higher Education 0075/DIA/2017/46 (WP) (www.mnisw.gov.pl The funders had no role in study design, data collection and analysis, decision to publish, or preparation of the manuscript.

**Competing interests:** The authors have declared that no competing interests exist.

advanced melanoma remains unsatisfactory [2–5]. Targeted therapy in melanoma depends on, among others, small-molecule inhibitors of the MAPK signalling pathway, that is overactivated in the majority of melanoma tumors [6]. Inhibitors of BRAF kinase (vemurafenib, dabrafenib, and encorafenib), as well as inhibitors of MEK kinase (trametinib, binimetinib, cobimetinib), were approved by the FDA. Vemurafenib was a first-in-class drug approved by the FDA in 2011 for the treatment of BRAF mutation-positive metastatic melanoma [7]. The randomized BRIM-3 study showed vemurafenib is associated with better survival rates comparing to conventional chemotherapeutic, dacarbazine [8]. Overall survival, in the study with the long follow up, was 16 months [9]. Despite the satisfactory response to vemurafenib, appearing resistance leads to the progression after 5.1–8.8 months. Melanoma cells gain resistance to vemurafenib *via*, i.e. reactivation of the MAPK and the PI3K–Akt pathway [10].

MicroRNAs (miRNAs, miRs) are single-stranded, stable, small non-coding RNAs which play an important role in post-transcriptional gene regulation by interacting with mRNAs, inhibiting their translation or leading to its degradation. They have pleiotropic effects because of targeting different mRNAs by single microRNA [11]. microRNAs play a significant role in the pathogenesis of different types of diseases, including cancers. Thus, they can serve as biomarkers in the diagnosis and prognosis of many diseases [12]. Recent studies show that microRNAs not only control key pathways leading to the development and progression of melanoma but also are capable of influencing the development of resistance to BRAF inhibitors [11, 13]. miR-204-5p and miR-211-5p are described to be upregulated by vemurafenib and both are stimulating MAPK pathways and also being involved in the emergence of melanoma cells resistance to BRAF inhibitor [14, 15].

miR-410-3p is a miRNA from the 14q32.2 mega-cluster which resides within Dlk-Dio3 domain associated with development and growth [16]. Recent studies demonstrated its role in numerous diseases including cardiomyopathy [17] and stroke [18]. It was described to play a significant role in the pathogenesis of different types of cancer as tumor suppressor miR (gastric, pancreatic, endometrial, breast and bone cancers) or oncomiR (liver, colorectal and non-small cell lung cancers) [19, 20]. miR-410-3p can promote or inhibit cell proliferation, invasion, migration and apoptosis through different factors. In our previous study, we found that miR-410-3p have a lineage-specific role in pituitary adenomas and controls crucial signalling pathways and modulates tumor cells invasiveness and proliferation [20]. miR-410-3p negatively modulates CETN3, BAK1, and BRD7 to stimulate oncogenesis and positively regulates AGTR1, c-MET and SNAIL leading to the suppressed cancer progression [19]. miR-410-3p influence chemosensitivity to gemcitabine in the treatment of pancreatic ductal adenocarcinoma by inhibiting HMGB1-mediated autophagy and thus may serve as a biomarker of chemoresistance [21].

Phenotype switching is a leading model explaining an aggressive behaviour and plasticity of melanoma [22]. Proliferative phenotype is characterized by high MITF expression and sensitivity to therapies. Invasive phenotype, conversely, is characterized by low MITF expression but high AXL level, as well as resistance to multiple therapies. Cells by switching from a proliferative to an invasive phenotype acquire resistance to treatment [23].

The aim of the study was to determine the role of miR-410-3p in the response and resistance to vemurafenib in melanoma.

## Materials and methods

### Patients tissue

The study was performed on archival, formalin-fixed, paraffin-embedded (FFPE) primary cutaneous melanoma tumors originating from 12 previously untreated patients. Patients data

are presented in S1 Table. The study was conducted in accordance with the Declaration of Helsinki. Study was approved by the Bioethical Committee Medical University of Warsaw (AKB/301/2019). FFPE samples were cut into 10 μm sections and mounted on glass slides (Super-Frost Ultra Plus, Menzel-Glazer, Braunschweig, Germany). Subsequently, standard HE staining was performed, and approximately 10 mm2 of tumor tissue, transient tissue, and healthy tissue were catapulted into separate tubes using Laser Capture Microdissection (Zeiss PALM MicroBeam, Germany). In cases of two patients (Mel006 and Mel008) adjacent tissues were not dissected. One patient (Mel005) had two distinct melanoma tumors. RNA was isolated using Norgen Biotek FFPE RNA/DNA Purification Plus Kit. Reverse transcription and real-time qPCR were performed as described above.

## Bioinformatical analysis

The expression of miR-410-3p in 30 types of cancer were analyzed using data from The Cancer Genome Atlas Research Network database using OncomiR [24]. The enrichment analysis of miR-410-3p targets in KEGG pathways as well as analysis of the correlation between miR-410-3p and mRNA expression was performed using starBase [25, 26]. P value for correlation was calculated using Pearson correlation test. Survival curves based on miR-410-3p expression were generated using OncoLnc [27]. Dlk-Dio3 domain was analyzed using the UCSC Genome Browser database [28]. A putative promoter sequence was analyzed using PROMO software [29].

## Cell culture

Experiments were performed on three human melanoma cell lines A375, G361, and SK-MEL1. Cell lines were kindly provided by prof. Maciej Małecki, who purchased them in ATCC. Cells were maintained according to the manufacturer's instructions in standard conditions (5% $CO_2$, 37˚C, humified atmosphere). All cell culture media and reagents were purchased from Gibco BRL (Gran Island, NY, USA). Cell lines were tested for mycoplasma contamination and were negative. All experiments were performed according to the Good Laboratory Practice.

## Vemurafenib and IC50

Vemurafenib (PLX4032) was purchased from Selleckchem (Catalog No. S1267, Batch no. 126712, purity 99.03%) and was dissolved in DMSO at the final concentration 1 mM and stored in -80˚C. IC50 was determined using the MTT method using CellTiter 96® Aqueous One Solution Cell Proliferation Assay (Promega) and was performed according to the manufacturer's protocol. $5 \times 10^3$ cells/well were seeded in 96-well plates and incubated with different doses of vemurafenib. After 48 h reagent was added, followed by 2h of incubation. Absorbance at 570 nm was measured using FLUOstar OPTIMA (BMG Labtech). IC50 was calculated using GraphPad Prism 8 (GraphPad Sofware Inc.). IC50s of cell lines are shown in S1 Fig. All experiments were performed in technical triplicates and were repeated at least in biological triplicates.

## RNA isolation, reverse transcription, and real-time qPCR

Total RNA from cells was isolated using the RNeasy Mini Kit (Qiagen). The quantity and purity of isolated RNA were assessed by the absorbance measurements at wavelengths of 260 nm and 280 nm on NanoDrop2000 spectrophotometer (Thermo Fisher Scientific Inc.). Samples with OD 260/280 ratios between 1.8 and 2.1 were used for further analysis. RNA was subjected to reverse transcription using Mir-X™ miRNA FirstStrand Synthesis followed by real-

time qPCR using SYBR® qRT-PCR (Takara, Clontech). Primers sequences used in the study: hsa-miR-410-3p: 5'-AATATAACACAGATGGCCTGT-3', AXL forward 5'- AACCTTCAAC TCCTGCCTTCT-3', reverse 5'-CAGCTTCTCCTTCAGCTCTTCAC-3', CHOP forward 5'-AGAACCAGGAAACGGAAACAGA-3', reverse 5'- TCTCCTTCATGCGCTGCTTT-3', ATF4 forward 5'- GTTCTCCAGCGACAAGGCTA-3', reverse 5'- ATCCTGCTTGCTGTTG TTGG-3', sXBP1 forward 5'-CTGAGTCCGAATCAGGTGCAG-3', reverse 5'- ATCCATG GGGAGATGTTCTGG-3', and GAPDH forward 5'- AGGGCTGCTTTTAACTCTGGT-3', reverse 5'-CCCCACTTGATTTTGGAGGGA-3' as an endogenous control for mRNA anlysis. U6 (Takara, Clontech) was used as an endogenous control for the analysis of microRNA expression. The mean Ct values of a target gene and endogenous control were used to calculate relative expression using the $2^{-\Delta Ct}$ method. For the calculation of relative expression of miR-410-3p in vemurafenib-treated cells, mean Ct values of a target gene and endogenous control (U6) were used in vemurafenib-treated cells and vehicle (DMSO)-treated cells using the $2^{-\Delta\Delta Ct}$ method.

## Transfection

All transfections were performed using jetPRIME (Polyplus) according to the manufacturer's protocol. miR-410-3p mimic (assay ID: MC11119), miR-scrambled (miR-scr, miRNA Mimic Negative Control), anti-miR-scrambled (anti-miR-scr, Anti-miR™ miRNA Inhibitor Negative Control), and anti-miR-410-3p (assay ID: AM11119) were obtained from Invitrogen™ mir-Vana™ (Thermo Fisher Scientific). miRs were used at a final concentration of 50 nM. Transfection efficiency was determined using real-time qPCR (S2 Fig) as was calculated in relative to miR-scrambled for mimic-miR-410-3p and anti-miR-scrambled for anti-miR-410-3p. To assess the role of miR-410-3p in response and resistance to vemurafenib, cells were seeded at 96-well plate and transfected with either mimic-miR, anti-miR, miR-scrambled, or anti-miR-scrambled. After 6h, vemurafenib was added to cells in final concentration equal to determined IC50. After 48h, resistance to vemurafenib after transfection was assessed using MTT method described above. Relative resistance was calculated as IC50 of mimic-miR-410-3p or anti-miR-410-3p transfected cells relative to corresponding miR-scr control.

## ER stress

Thapsigargin was purchased from Sigma-Aldrich, Inc. (Merck). Cells were treated with 60 nM thapsigargin (TG) for 48h, followed by RNA isolation, reverse transcription and real-time qPCR, as described above. The induction of ER stress was confirmed by real-time qPCR. To test the level of miR-410-3p after treatment with vemurafenib and TG, cells were incubated with TG (60 nM) and vemurafenib (IC50) for 24h.

## Statistical analysis and data presentation

All experiments were performed at least in triplicate. Data distribution was tested using Shapiro-Wilk test. Appropriate statistical tests were applied to assess mean differences between groups, paired t test, Wilcoxon matched-pairs signed rank test. All statistical tests were performed using GraphPad Prism 8 (GraphPad Sofware Inc.). All values are represented as mean ± SD. A p-value of <0.05 was considered statistically significant.

## Results

### miR-410-3p is downregulated in untreated melanoma tumors

miR-410-3p has a divergent role in molecular oncology and may act as either oncomiR or tumor suppressor miR. We analyzed the expression of miR-410-3p in a panel of 30 types of

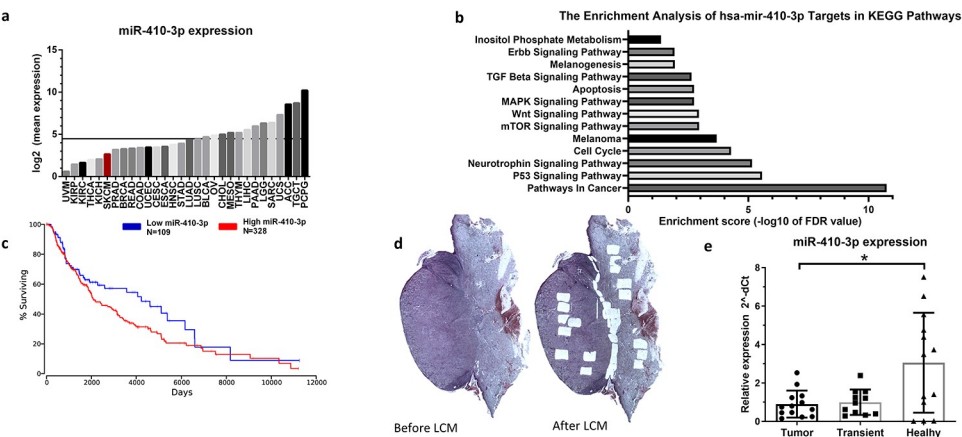

**Fig 1. miR-410-3p regulates multiple pathways in cancer and is downregulated in melanoma tumors. (a)** The expression of miR-410-3p in different types of cancer based on TCGA [24] Grey line–mean expression for all types of cancer **(b)** The enrichment analysis of miR-410-3p targets in KEGG pathways based on TCGA using starBase [25, 26] **(c)** Kaplan plot for mir-410-3p in melanoma based on TCGA survival data using OncoLnc [27]. log-rank p-value = 0.0764 **(d)** Tumor scans before and after Laser Capture Microdissection (LCM) **(e)** The expression of miR-410-3p is downregulated in tumor and transient tissues compared with adjacent healthy skin tissues. Wilcoxon matched-pairs signed rank test. *—p<0.05. Abbreviations: ACC—Adrenocortical carcinoma, BLCA—Bladder urothelial carcinoma, BRCA—Breast invasive carcinoma, CESC—Cervical and endocervical cancers, CHOL—Cholangiocarcinoma, COAD—Colon adenocarcinoma, HNSC—Head and Neck squamous cell carcinoma, KICH—Kidney Chromophobe, KIRC—Kidney renal clear cell carcinoma, KIRP—Kidney renal papillary cell, LGG—Brain Lower Grade Glioma, LIHC—Liver hepatocellular carcinoma, LUAD—Lung adenocarcinoma, LUSC—Lung squamous cell carcinoma, MESO—Mesothelioma, OV—Ovarian serous cystadenocarcinoma, PAAD—Pancreatic adenocarcinoma, PCPG—Pheochromocytoma and Paraganglioma, PRAD—Prostate adenocarcinoma, READ—Rectum adenocarcinoma, SARC—Sarcoma, SKCM—Skin Cutaneous Melanoma, STAD—Stomach adenocarcinoma, TGCT—Testicular Germ Cell Tumors, THCA—Thyroid carcinoma, THYM—Thymoma, UCEC—Uterine Corpus Endometrial Carcinoma, UCS—Uterine Carcinosarcoma, UVM—Uveal Melanoma.

cancer from The Cancer Genome Atlas (TCGA) Research Network database using OncomiR [24]. We found that the level of miR-410-3p was lower in melanomas (mean expression = 2.64) compared with the mean for all types of cancer (mean expression = 4.48 ± 2.23, Fig 1a). Moreover, we performed the enrichment analysis of miR-410-3p targets in KEGG pathways using StarBase [25, 26]. It identified several signaling pathways related to cancer, including melanoma, as regulated by miR-410-3p-target axis (Fig 1b, S1 Table). Moreover, we analyzed the TCGA survival data using OncoLnc [27]. We found that there is a slight association between higher level of miR-410-3p and shorter overall survival (Fig 1c). The difference is the most prominent during the first 10 years. To accurately determine the level of miR-410-3p in melanoma, we checked the expression of miR-410-3p in tumor tissues, transient tissues, and adjacent healthy skin dissected from 12 FFPE (formalin-fixed, paraffin-embedded) primary nodular melanoma (Fig 1d). We found that the expression of miR-410-3p was downregulated in tumor tissues compared with corresponding healthy skin tissues (Fig 1d).

## miR-410-3p is induced by vemurafenib

For *in vitro* study we used three model human melanoma cells lines, i.e., A375 (homozygous BRAF V600E), G361 (heterozygous BRAF V600E), and SKMEL1 (heterozygous BRAF V600E). First, we determined the IC50 of vemurafenib for 48h of cells incubation with the drug. Next, we incubated melanoma cells with vemurafenib concentration equal to IC50 (A375–98.56 nM, G361–401.35 nM, SKMEL1–276.59 nM) followed by RNA isolation after 24h, 48h, and 96h. We found that the expression of miR-410-3p was significantly induced 48h and 96h after vemurafenib was administered in melanoma cell lines (Fig 2a). It suggested the

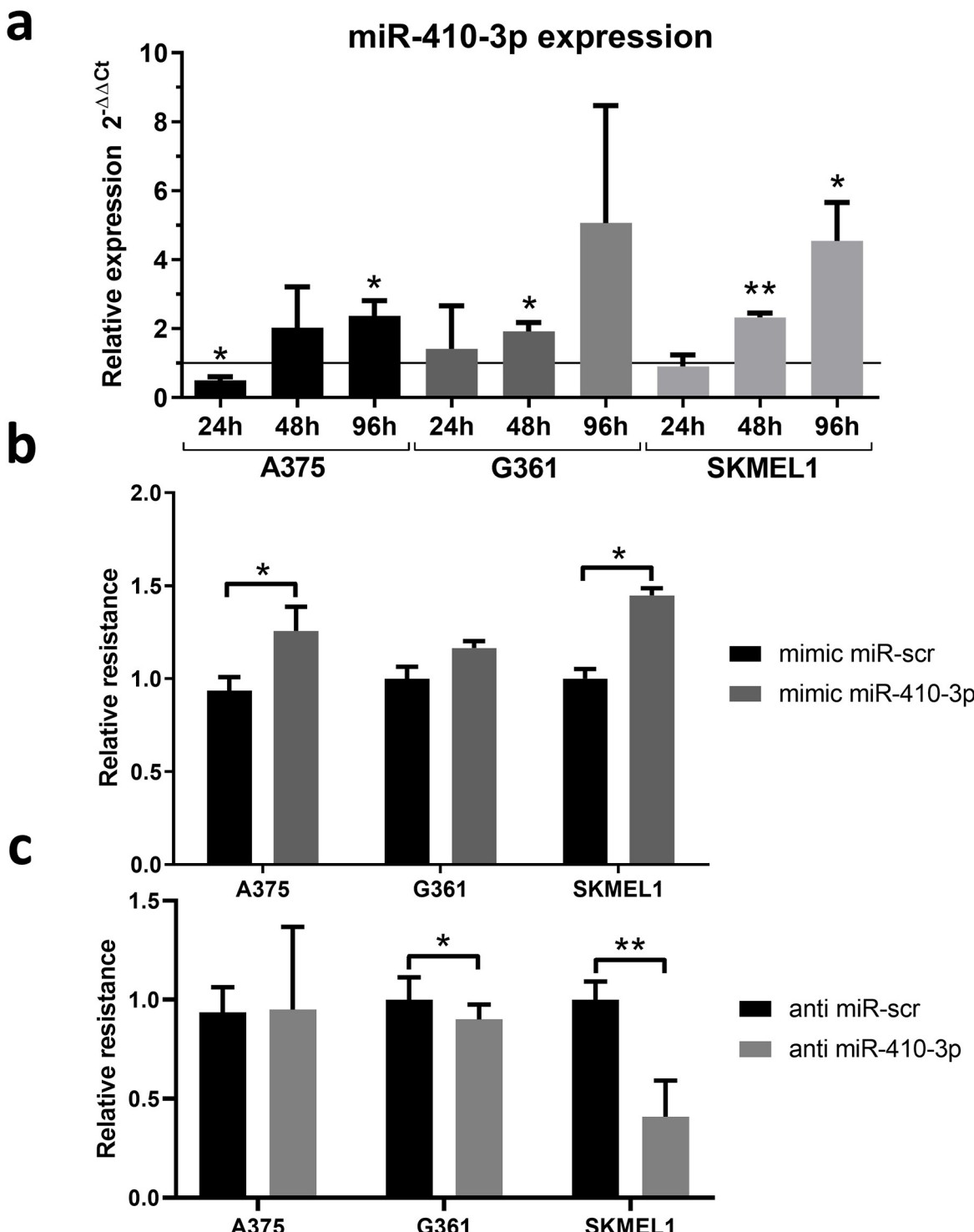

**Fig 2. miR-410-3p is induced by vemurafenib and leads to the vemurafenib resistance in melanoma cell lines. (a)** The induction of the expression of miR-410-3p after 24h, 48h, and 96h of incubation with vemurafenib. miR-410-3p expression is presented as relative to the vehicle (DMSO) control. **(b)** Mimic miR-410-3p increased the resistance to vemurafenib **(c)** Inhibition of miR-410-3p by anti-miR-410-3p sensitized melanoma cells to vemurafenib. Paired t test *—p<0.05, **—p<0.01. Relative resistance was calculated as IC50 of mimic-miR-410-3p or anti-miR-410-3p transfected cells relative to corresponding miR-scr control.

role of miR-410-3p in either response to vemurafenib by melanoma cells or as the mechanism of vemurafenib action.

## miR-410-3p increases melanoma cells resistance to vemurafenib

In order to assess the role of miR-410-3p in response and resistance to vemurafenib, we performed the MTT assay and calculated the IC50 after transfection with either synthetic mimic miR-410-3p or anti miR-410-3p. We found that mimic miR-410-3p increased melanoma cells resistance to vemurafenib (Fig 2b), while inhibition of miR-410-3p led to decrease of resistance (Fig 2c). It showed that miR-410-3p is one of the mechanisms of response and resistance to BRAFi occurring in melanoma cells.

## The expression of miR-410-3p is induced by ER stress

miR-410-3p is a member of mega-cluster Gtl2-Dio3 [30]. It is suggested that the whole mega-cluster is coordinately regulated and expressed [31]. We analyzed *in silico* Dlk-Dio3 domain using the UCSC Genome Browser database and H3K4Me3 mark [28]. We analyzed the putative promoter sequence that was located in the first exon and intron of MEG3 gene (Fig 3a) using PROMO software [29]. We have found that many transcription factors

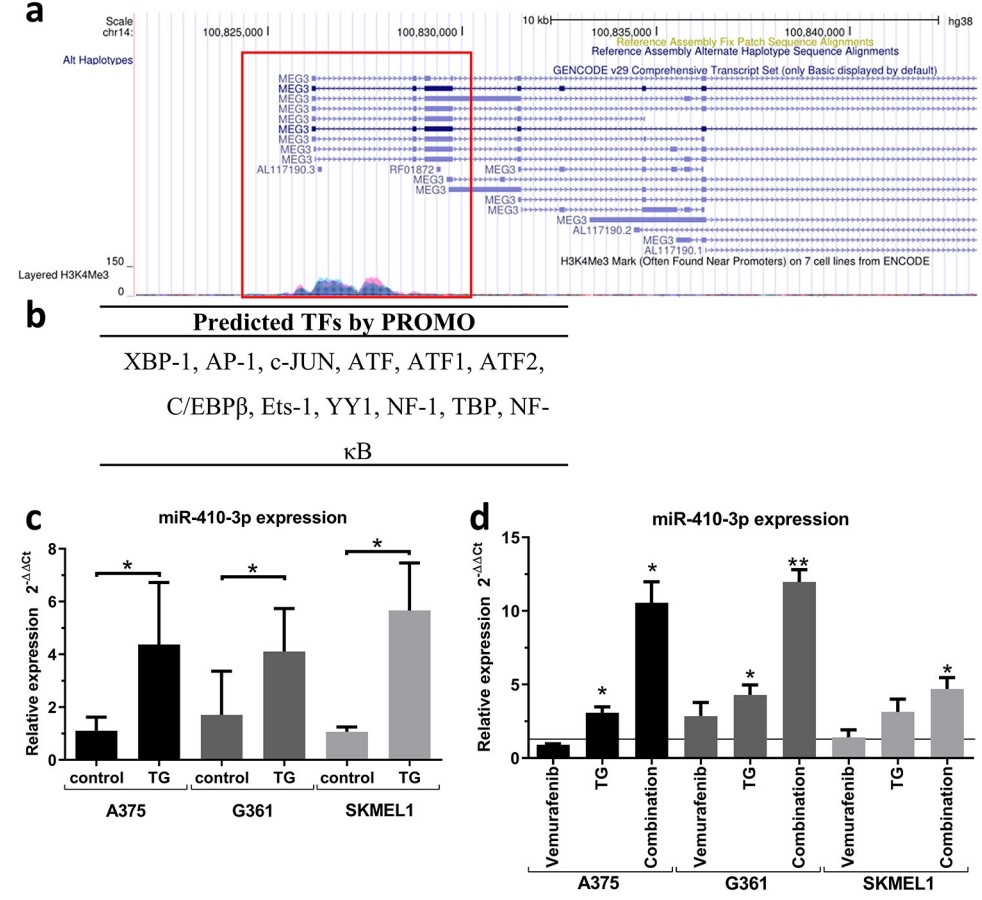

**Fig 3. miR-410-3p is induced by vemurafenib via ER stress. (a)** In silico analysis of Dlk-Dio3 domain using UCSC Genome Browser database and H3K4Me3 mark. **(b)** Transcription factors predicted to bind to putative promoter sequence using PROMO software. **(c)** ER stress induction using thapsigargin (TG) upregulated the expression of miR-410-3p. **(d)** ER stress induction by TG increases vemurafenib-induced miR-410-3p expression. Paired t test *—p<0.05. **—p<0.01.

associated with ER stress response are predicted to target promoter sequence (Fig 3b). More-over, Kato et al. found that the part of mega-cluster that includes miR-410-3p, is controlled by CHOP, a multifunctional transcription factor in the ER stress response [32]. Beck et al. found that vemurafenib induces ER stress in melanoma cells [3]. Targeting ER stress-induced autophagy is a promising way to overcome BRAF inhibitor resistance [33]. We found that in our setting vemurafenib induced ER stress (S3 Fig). Therefore, we checked whether the induction of miR-410-3p expression is mediated by ER stress. We induced ER stress in melanoma cells using thapsigargin, a non-competitive inhibitor of the endoplasmic reticulum calcium ATPase, TG (S4 Fig). We found that ER stress significantly upregulated the level of miR-410-3p in all three melanoma cell lines (Fig 3c). Combined treatment with both vemurafenib and TG additively enhanced the induction of miR-410-3p in melanoma cells (Fig 3d). Therefore, induction of the expression of miR-410-3p by vemurafenib is at least partially mediated by the ER stress.

### miR-410-3p favors melanoma switching toward invasive, therapy-resistant phenotype

Melanoma cells are characterized by high plasticity and capacity to switch between invasive and proliferative phenotypes, which is one of the reasons for a remarkable tumor heterogeneity [22, 34, 35]. MAPK inhibitors induce switching from the proliferative to the invasive pheno-type, that leads to the resistance [23]. We analyzed the correlation between the expression of miR-410-3p and markers of either proliferative or invasive phenotype in 449 skin cutaneous malignant melanoma from starBase [25, 26]. We found that the expression of miR-410-3p cor-relates with the expression of AXL (Fig 4a), a regulator of invasive phenotype [36]. Therefore, we checked whether miR-410-3p affects phenotype switching *in vitro*. We found that miR-410-3p promotes phenotype switching in A375 cell line toward the invasive phenotype, based on the expression of AXL (Fig 4b). In a more detailed analysis of starBase, we found that miR-410-3p negatively correlates with several markers of the proliferative phenotype (Fig 4c). Con-versely, the expression of miR-410-3p and markers of invasive phenotype were positively cor-related (Fig 4a and 4d).

## Discussion

Our study showed a novel mechanism of vemurafenib response in which miR-410-3p is induced by vemurafenib *via* ER stress, which may contribute to the phenotype switching toward therapy-resistant phenotype (Fig 5).

We showed that the level of miR-410-3p is lower in melanoma tissues samples compart-ments compared to adjacent healthy skin tissues from untreated melanoma patients. More-over, its level is low compared to other types of cancer, based on TCGA. Importantly, the majority of tumors from TCGA were naive (untreated) [37], which may explain the miR-410-3p level in that samples. Interestingly, high level of miR-410-3p in melanoma tumors is associ-ated with slightly poorer prognosis. We suggest that despite low expression in untreated tumors, miR-410-3p is induced after vemurafenib treatment and contributes to the resistance. Since miR-410-3p may induce the phenotype switching toward the invasive phenotype, our results are in line with the results of Muller et al. who showed that low MITF/AXL ratio pre-dicts early resistance to multiple targeted drugs in melanoma [36].

We performed the enrichment analysis of miR-410-3p targets in KEGG pathways based on TCGA using starBase. miR-410-3p-targets axis was found to be associated with cancer signal-ing and BRAFi resistance pathways, including MAPK, mTOR [38], and neurotrophins [39] microRNAs contribute to the development and progression of melanoma [11]. It is suggested

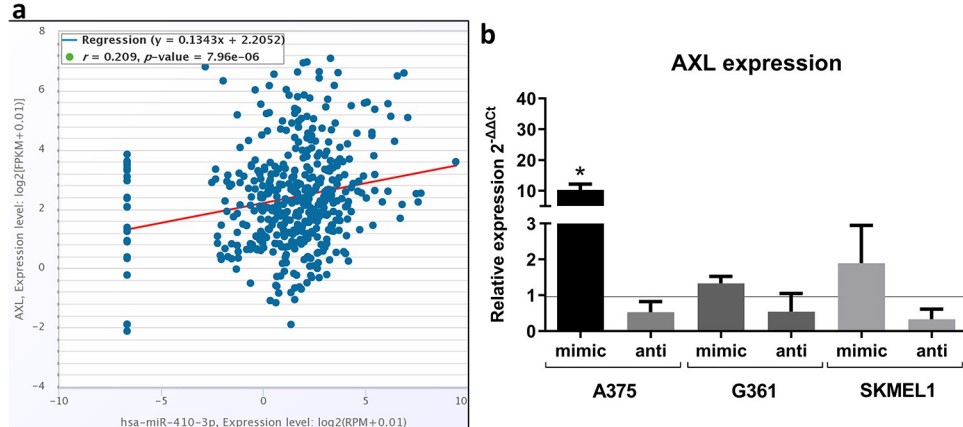

| c Gene | r | P-value |
|--------|--------|----------|
| MITF | -0.205 | 1.14e-05 |
| MLANA | -0.186 | 7.41e-05 |
| DUSP2 | -0.093 | 4.86e-02 |
| CDK2 | -0.080 | 9.06e-02 |
| CAPN3 | -0.216 | 3.98e-06 |
| ASAH1 | -0.128 | 6.66e-03 |
| APOE | -0.088 | 6.16e-02 |

| d Gene | r | P-value |
|--------|--------|----------|
| NTM | 0.365 | 1.45e-15 |
| TLE4 | 0.085 | 7.20e-02 |
| TPM1 | 0.273 | 3.84e-09 |
| WNT5A | 0.240 | 2.60e-07 |
| ZEB1 | 0.204 | 1.38e-05 |
| EGFR | 0.265 | 1.12e-08 |
| OSMR | 0.217 | 3.45e-06 |

**Fig 4. miR-410-3p correlates with markers of invasive phenotype and favors melanoma switching toward invasive, therapy-resistant phenotype [25, 26]. (a)** The correlation between miR-410-3p expression and AXL in melanoma. **(b)** Upregulation of miR-410-3p after transfection upregulated the expression of AXL. Paired t test **(c)** Correlation between the level of miR-410-3p and markers of proliferative phenotype [25, 26]. **(d)** Correlation between the level of miR-410-3p and markers of the invasive phenotype [25, 26].

that microRNAs deregulation is a major non-genomic alteration that drives melanoma resistance [40]. The most relevant intracellular pathways affected by deregulated miRNAs during vemurafenib treatment include inflammation, angiogenesis, MAPK signaling, as well as cell cycle and apoptosis [40]. miR-410-3p regulates multiple pathways in the cell including MAPK, Wnt, mTOR and p53. It was shown that miR-204-5p and miR-211-5p contribute to the BRAF inhibitor resistance [14]. Loss of miR-211 sensitizes melanoma cells to vemurafenib treatment [15]. However, miR-204-5p and miR-211-5p were upregulated also by trametinib (MEK inhibitor) and SCH772984 (ERK inhibitor), whereas their levels remain stable in response to AKT inhibitor or Rac inhibitors [14]. It suggested that the induction of these microRNAs is a universal mechanism of response to the inhibition of whole MAPK signaling pathway. Importantly, these studies focused on vemurafenib-resistant cell lines or used high-dose vemurafenib. In our study, we focused on an early response of melanoma cells to vemurafenib. Another miRNA involved in the regulation of melanoma response and resistance to vemurafenib is miR-514a that inhibits NF1 expression and thus confers vemurafenib resistance [41]. miR-514a is highly upregulated in melanoma vemurafenib resistant cells [42]. Moreover, it was shown that miR-34a, miR-100, and miR-125b are upregulated in vemurafenib-resistant cell lines and in tumors obtained from patients treated with BRAFi [43]. Conversely, miR-579-3p, melanoma tumor suppressor miR, is downregulated in vemurafenib-resistant cells and BRAF inhibitor-resistant patients [44]. Likewise, miR-7 was found to be strongly downregulated in

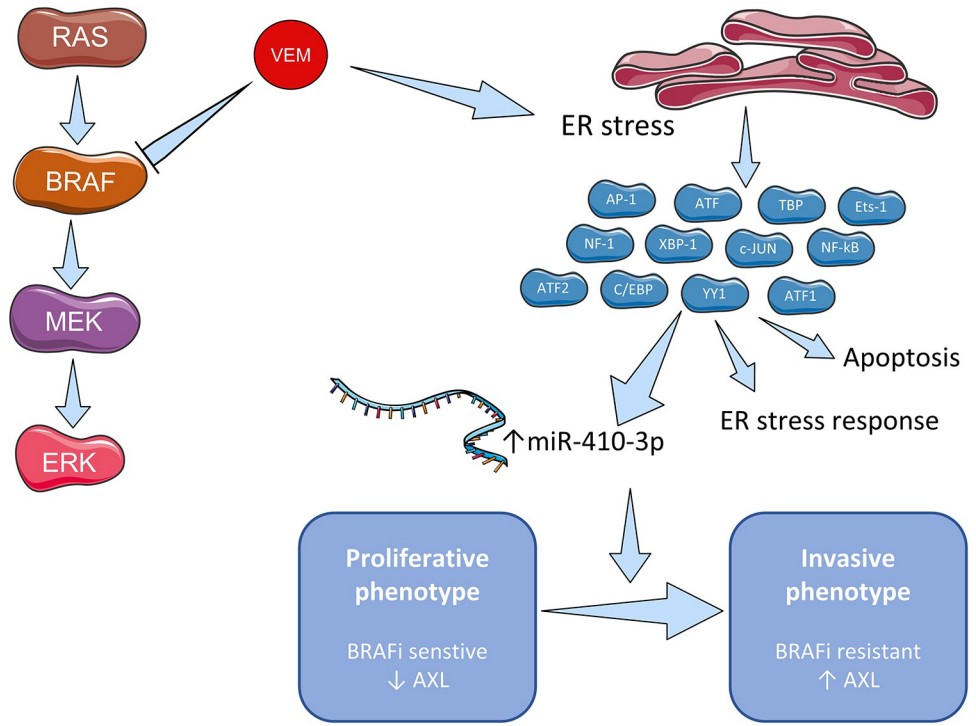

**Fig 5. The role of microRNAs in response and resistance to vemurafenib in melanoma cells.** We described a novel mechanism of initial resistance response to vemurafenib via miR-410-3p. Vemurafenib inhibits MAPK signaling pathway as well as induces ER stress that leads to the expression of miR-410-3p. It favors switch towards invasive, BRAFi resistant phenotype. The figure was prepared using Servier Medical Art (https://smart.servier.com/).

vemurafenib resistant cells due to its role as the suppressor of MAPK and PI3K/AKT pathways [42].

Vemurafenib also upregulates miR-524-5p, however, its role is opposed to the aforementioned microRNAs [45]. Its expression is downregulated in activated MAPK signaling and is induced by BRAFi or MEK1/2i. miR-524-5p suppress cell proliferation, growth, and slows the progression of melanoma in mice [45]. Therefore, it potentiates the inhibition of MAPK signaling by vemurafenib *via* targeting BRAF and ERK2. Moreover, several microRNAs were identified as regulated by RAF/MEK/ERK signaling pathway in melanoma [46] and its expression may be deregulated by MAPK inhibition. That confirms the sophisticated role of microRNA in the response and resistance to vemurafenib. The complexity of microRNAs network in melanoma remains unclear, especially in the context of targeted therapy.

We found that miR-410-3p is induced by vemurafenib in melanoma cells. The enrichment analysis of miR-410-3p suggested its role in the regulation of crucial signalling pathways in cancer, including melanoma. In order to understand the mechanism of miR-410-3p induction, we analyzed the promoter and regulatory sequences of miR mega-cluster Dlk-Dio3. We identified several transcription factors using PROMO software [29], and 12 of them are related to the regulation of ER stress response. ER stress activates c-JUN amino-terminal kinases (JNKs) [47], which in turn enable interaction with JunB, JunD, c-Fos, and ATF that constitute the AP-1 transcription factor [48]. ER stress triggers IRE1α dimerization, followed by autophosphorylation and a conformational shift. It activates its C-terminal endoribonuclease domain to cleave 26 nucleotides from the Xbp1 mRNA, followed by re-ligation by the tRNA ligase RTCB [49], that enables the translation of the functionally active XBP1, that regulates the ER stress

response. Moreover, ER stress induces the expression of the transcription factor C/EBP-β [50] and Ets-1 [51]. ER stress induces the expression of Grp78, a prosurvival ER chaperone, via YY1 that is a multifunctional transcription factor [52]. Other transcription factors that control the ER stress or vemurafenib response and that are predicted to bind to the promotor include TBP, NF-1, NF-κB, and CREB [53]. Since it was shown that vemurafenib induces ER stress [3], we checked whether vemurafenib induces the expression of miR-410-3p *via* ER stress. We demonstrated that the induction of ER stress by thapsigargin upregulated the expression of miR-410-3p. It suggested that vemurafenib induces the expression of miR-410-3p *via* ER stress-related transcription factors.

Upregulation of miR-410-3p, similarly to miR-204-5p and miR-211-5p, led to the increased resistance to vemurafenib in A375 and SKMEL1 cells, while inhibition of miR-410-3p sensitizes G361 and SKMEL1 melanoma cells to this drug. It confirmed that the induction of miR-410-3p is a mechanism of melanoma cells response to vemurafenib. Lack of the observed effect in all tested cell lines may results from differences in endogenous expression of miR-410-3p in melanoma cells.

Phenotype switching is a leading model of complex melanoma behavior. The switch from proliferative to invasive phenotype is one of the mechanisms of the response and early resistance to stress factors, including targeted therapy [22, 54]. The phenotype switching phenomenon is similar to the epithelial-mesenchymal transition in epithelial cancers [55]. We have found that miR-410-3p expression correlates with the expression of the markers of the invasive phenotype. Moreover, miR-410-3p induced the expression of AXL in A375, one of the main regulators of the invasive phenotype, and thus contributing to the vemurafenib resistance. Further research are required to dissect the exact role of miR-410-3p in the regulation of vemurafenib response and resistance.

## Conclusions

In this paper, we described a comprehensive mechanism by which melanoma cells acquire resistance to vemurafenib. miR-410-3p is induced by vemurafenib in melanoma cells possibly by induction of ER stress. It leads to the switch toward invasive, therapy-resistant phenotype and eventually contributes to the resistance to BRAF inhibitors.

## Supporting information

**S1 Fig. IC50 of vemurafenib in studied cell lines.**
(TIFF)

**S2 Fig. Transfection efficiency.** The efficiency of the transfection was determined using qPCR. The expression of miR-410-3p is presented as relative expression compared to the miR-scrambled for mimic-miR-410-3p and anti-miR-scrambled for anti-miR-410-3p.
(TIFF)

**S3 Fig. Expression of ER stress markers in vemurafenib-treated melanoma cells.** The expression of ER stress markers are presented as relative expression compared to vehicle (DMSO)-treated cells. *—p<0.05, **—p<0.01, ***—p<0.001, ****—p<0.0001.
(TIFF)

**S4 Fig. Expression of ER stress markers in TG-treated melanoma cells.** The expression of ER stress markers, CHOP, ATF4 and sXBP1 was determined using qPCR. *—p<0.05.
(TIFF)

**S1 Table. Clinical data of the patients involved in the study.**
(DOCX)

**S2 Table. Results from the enrichment analysis of miR-410-3p targets in KEGG pathways using starBase.**
(DOCX)

## Acknowledgments

We would like to offer our thanks and appreciation for the insightful discussions to the members of Department of Immunology, Medical University of Warsaw. The results published here are in part based upon data generated by the TCGA Research Network: https://www.cancer.gov/tcga.

## Author Contributions

**Conceptualization:** Tomasz M. Grzywa, Klaudia Klicka.

**Data curation:** Tomasz M. Grzywa, Klaudia Klicka.

**Formal analysis:** Tomasz M. Grzywa, Klaudia Klicka.

**Funding acquisition:** Tomasz M. Grzywa, Wiktor Paskal.

**Investigation:** Tomasz M. Grzywa, Klaudia Klicka, Wiktor Paskal, Julia Dudkiewicz.

**Methodology:** Tomasz M. Grzywa, Klaudia Klicka.

**Project administration:** Tomasz M. Grzywa, Paweł K. Włodarski.

**Resources:** Tomasz M. Grzywa, Klaudia Klicka, Wiktor Paskal, Jarosław Wejman, Michał Pyzlak, Paweł K. Włodarski.

**Software:** Tomasz M. Grzywa.

**Supervision:** Tomasz M. Grzywa.

**Validation:** Tomasz M. Grzywa, Klaudia Klicka, Wiktor Paskal.

**Visualization:** Tomasz M. Grzywa, Klaudia Klicka.

**Writing – original draft:** Tomasz M. Grzywa, Klaudia Klicka.

**Writing – review & editing:** Tomasz M. Grzywa, Klaudia Klicka, Wiktor Paskal, Julia Dudkiewicz, Jarosław Wejman, Michał Pyzlak, Paweł K. Włodarski.

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
