## [Decision Letter · Decision Letter 0]

28 Apr 2020

PONE-D-20-03181

miR-410-3p is induced by vemurafenib via ER stress and contributes to resistance to BRAF inhibitor in melanoma

PLOS ONE

Dear Dr Paskal,

Thank you for submitting your manuscript to PLOS ONE. After careful consideration, we feel that it has merit but does not fully meet PLOS ONE’s publication criteria as it currently stands. Therefore, we invite you to submit a revised version of the manuscript that addresses the points raised during the review process.

A number of issues were raised by the reviewers, particularly Reviewer 1 which should be addressed if the authors plan to submit a revised manuscript.

We would appreciate receiving your revised manuscript by Jun 12 2020 11:59PM. To enhance the reproducibility of your results, we recommend that if applicable you deposit your laboratory protocols in protocols.io, where a protocol can be assigned its own identifier (DOI) such that it can be cited independently in the future. For instructions see: http://journals.plos.org/plosone/s/submission-guidelines#loc-laboratory-protocols

We look forward to receiving your revised manuscript.

Kind regards,

Salvatore V Pizzo

Academic Editor

PLOS ONE

Journal Requirements:

2. Please provide additional information about each of the cell lines used in this work, including history, specific culture conditions and any quality control testing procedures (authentication, characterisation, and mycoplasma testing).

For more information, please see http://journals.plos.org/plosone/s/submission-guidelines#loc-cell-lines.

3. At this time, we ask that you please provide the product number and any lot numbers of the Vemurafenib inhibitor purchased from Selleckchem used in this study.

4. To comply with PLOS ONE submission guidelines, in your Methods section, please provide additional information regarding your statistical analyses.

For more information on PLOS ONE's expectations for statistical reporting, please see https://journals.plos.org/plosone/s/submission-guidelines.#loc-statistical-reporting.

5. Your ethics statement must appear in the Methods section of your manuscript. If your ethics statement is written in any section besides the Methods, please move it to the Methods section and delete it from any other section. Please also ensure that your ethics statement is included in your manuscript, as the ethics section of your online submission will not be published alongside your manuscript.

Reviewers' comments:

Reviewer's Responses to Questions

**Comments to the Author**

1. Is the manuscript technically sound, and do the data support the conclusions?

Reviewer #1: Partly

Reviewer #2: Yes

2. Has the statistical analysis been performed appropriately and rigorously? 

Reviewer #1: Yes

Reviewer #2: Yes

3. Have the authors made all data underlying the findings in their manuscript fully available?

Reviewer #1: Yes

Reviewer #2: Yes

4. Is the manuscript presented in an intelligible fashion and written in standard English?

Reviewer #1: Yes

Reviewer #2: Yes

5. Review Comments to the Author

Reviewer #1: SUMMARY OF RESEARCH AND OVERALL IMPRESSION OF THE MANUSCRIPT

The study conducted by Grzywa et al contributes to the field of microRNAs and their role in resistance to targeted therapy in melanoma with the finding of a novel miRNA that could be involved in this process. miR-410-3p could play a role in cancer, either as an oncomiR or as a tumor suppressor miR, as its expression differs across different cancer types using the TCGA database as well as the bioinformatics analysis showed that its predicted targets may be involved in cancer pathways. Furthermore, the expression of miR-410-3p in the melanoma patients samples analysed in the current study is lower in the tumours samples compared to the healthy portions.

Through different experiments the authors showed that miR-410-3p expression is upregulated by vemurafenib treatment in sensitive cells, which is more clearly detected after 48h of treatment. Either the transient overexpression or inhibition of miR-410-3p resulted in an slightly increase of resistance to vemurafenib in some of the human melanoma cell lines tested. Moreover, the increase of miR-410-3p expression is also achieved after treatment with thapsigargin, which the authors linked to ER stress activation. Also, AXL expression is increased in A375 melanoma cell line when miR-410-3p is overexpressed.

In overall, the authors described the role that miR-410-3p could have in resistance to vemurafenib, but a substantial amount of controls are missing, that should be included in the study. Also, some of the conclusions raised by the authors should be taken cautiously and should be expressed in a different way (they could be presented as hypotheses as the results for some experiments may not be clear in all the cell lines used).

After addressing the points mentioned the manuscript could be a good option to be published in Plos One.

Mayor points

Figure 2: The results of the experiments performed to determine the IC50 of vemurafenib for 48h with the different human melanoma cell lines should be included.

Figure 2a: miR-410-3p expression in cell lines treated with the vehicle should be included.

Figure 2b-c: The increase in resistance to vemurafenib after overexpression or inhibition of miR-410-3p is not seem in all the cell lines tested, this can not be ignored and should be mentioned in the text.

Figure 2: Transient transfection efficiency should be shown: if the experiments are performed 48h after transfection, results showing changes in miR-410-3p expression, either its increase with the mimic or its inhibition with the anti-miRNA compared to the corresponding controls (miR-scr or anti-miR-scr, respectively), should be included at this time point.

Figure 3c: It would strengthen the results if the expression of some ER stress markers were analyzed by qPCR to prove that ER stress had been successfully achieved with the conditions used in relation to thapsigargin.

Figure 3c: it would be interesting to perform an experiment with one of the cell lines comparing the expression of miR-410-3p in response to vemurafenib, thapsigargin and the combination of both.

Figure 4b: changes in AXL expression are only statistically significant after overexpressing miR-410-3p in the A375 cell line, this should be mentioned in the text.

Figure 4b: the expression levels of AXL in the transfectant controls are missing (miR-scrambled and anti-miR-scrambled).

Figure 5 (upper left): Most of miRNAs expression changes show here seem to be modified after BRAF inhibitor treatments. Apparently, by what is shown in the figure and how it is designed, no differences in most of miRNA expression occurs after treatment with MEK and ERK inhibitors. This is inaccurate, as for example, miR-410-3p expression in response to MEK and ERK inhibitors has not been analysed in this study, which raises the possibility that the same is happening in the others studies mentioned here. This information is relevant and it must be included in the scheme for all the miRNAs reported according to the studies used to create this figure.

Figure 5 (bottom right): The upregulation of miR-410-3p expression can not be related to a downregulation of MITF expression as this issue has not been addressed anywhere in this study, remove it or perform the corresponding experiments to claim so.

In general, the writing quality of the manuscript could be improved, and authors should revised it.

Minor points

Figure 1a: the meaning of the abbreviations for the types of cancer analysed are missing.

Figure 1b: the second highest enrichment pathway for miR-410-3p targets is the p53 signalling pathway. It would be of interest to include this finding in the discussion, along with any hypothesis in relation with the present study.

Figure 1c-line 169: as the differences in survival (Figure 1c) does not seem to be significant (p=0.0764), the sentence addressing this result could be less strong (such as “there is a slight association”). Also, the period of time when this difference is higher could be specifically mention in the text.

Line 176: the BRAF mutation status of the cell lines used should be mentioned.

Figure 4a-methods: the statistical method used to analyze the correlation of AXL and miR-410-3p expression must be included in the methods section.

Figure 4: the results about the switching towards a more invasive phenotype due to miR-410-3p expression would be strengthened if any marker of the proliferative phenotype would be characterised when overexpressing or inhibiting this miRNA.

Discussion: the low miR-410-3p expression found in the tumours sections of melanoma patients samples (Figure 1d) should be further discussed in relation to its significance in the context of the upregulation of this miRNA after vemurafenib treatment in human melanoma cell lines. Also, if the patients included in this study were treated or not should be clearly stated.

Reviewer #2: The manuscript by Paskal and coauthors reports about the involvement of miR410 in melanoma resistance to BRAF targeted drugs. The authors show that miR410 is induced by the drug in cell lines and its inhibition or ectopic expression associates with in vitro drug sensitivity. Analysis in a set of melanoma primary tumors showed that miR410 expression is lower in melanoma cells compared to adjacent skin tissue, and in silico analysis showed that miR410 expression correlates with markers associated to drug resistance.

The study is clearly written and reports interesting results, adding miR410 to previously reported miRs associated to melanoma resistance to the effect of BRAF targeted drugs. Nonetheless, I have few points that need clarification.

- The authors should explain how they identified miR410 for their study

- Please explain more about the importance of mir410 and its mechanism in the introduction section (according to previous studies)

- Table 1 reports clinical data about the studied tumors which are not of interest in the results section, and should be reported as supplementary information

- Fig 5 should be better focused on miR410 effects and mechanism

- Supplementary table 1 lacks a title and an explanation, and why some lines are in bold

- Fig 3a and 4a are poorly readable

- The authors should include the control gene used for miR410 level analysis by qPCR (fig 1e).

6. PLOS authors have the option to publish the peer review history of their article (what does this mean?). If published, this will include your full peer review and any attached files.

Reviewer #1: No

Reviewer #2: No

---

## [Author Response · Author response to Decision Letter 0]

18 May 2020

Dear Dr Salvatore V Pizzo, Academic Editor, PLOS ONE

We have submitted our revised manuscript "miR-410-3p is induced by vemurafenib via ER stress and contributes to resistance to BRAF inhibitor in melanoma". We have addressed all the points raised during the review process. We have modified our manuscript according to the PLOS ONE's style requirements, and we have added additional information about cell lines, vemurafenib batch and purity and statistical analysis. We have also added ethics statement to the Methods section. A rebuttal letter that responds to each point is attached as "Response to Reviewers".

Kind regards,

Wiktor Paskal

Department of Methodology

Medical University of Warsaw

RESPONSE TO REVIEWERS

Reviewer #1

The study conducted by Grzywa et al contributes to the field of microRNAs and their role in resistance to targeted therapy in melanoma with the finding of a novel miRNA that could be involved in this process. miR-410-3p could play a role in cancer, either as an oncomiR or as a tumor suppressor miR, as its expression differs across different cancer types using the TCGA database as well as the bioinformatics analysis showed that its predicted targets may be involved in cancer pathways. Furthermore, the expression of miR-410-3p in the melanoma patients samples analysed in the current study is lower in the tumours samples compared to the healthy portions.

Through different experiments the authors showed that miR-410-3p expression is upregulated by vemurafenib treatment in sensitive cells, which is more clearly detected after 48h of treatment. Either the transient overexpression or inhibition of miR-410-3p resulted in an slightly increase of resistance to vemurafenib in some of the human melanoma cell lines tested. Moreover, the increase of miR-410-3p expression is also achieved after treatment with thapsigargin, which the authors linked to ER stress activation. Also, AXL expression is increased in A375 melanoma cell line when miR-410-3p is overexpressed.

In overall, the authors described the role that miR-410-3p could have in resistance to vemurafenib, but a substantial amount of controls are missing, that should be included in the study. Also, some of the conclusions raised by the authors should be taken cautiously and should be expressed in a different way (they could be presented as hypotheses as the results for some experiments may not be clear in all the cell lines used).

Authors’ response:

We thank Reviewer for appreciation our study. We corrected our manuscript and performed additional experiments and we hope that our article will be suitable for publication in revised version. 

Figure 2: The results of the experiments performed to determine the IC50 of vemurafenib for 48h with the different human melanoma cell lines should be included.

Authors’ response: 

We included the results of IC50 of vemurafenib as S1 Fig. 

Figure 2a: miR-410-3p expression in cell lines treated with the vehicle should be included.

Authors’ response:

Figure 2a shows the relative expression of miR-410-3p in vemurafenib-treated cells compared to vehicle (DMSO). miR-410-3p were calculated using 2-ΔΔCt method as:

Relative expression = 2 ΔCt (Ct of miR-410-3p – Ct of endogenous control (U6) of vemurafenib-treated cells) / 2 ΔCt (Ct of miR-410-3p – Ct of endogenous control (U6) of DMSO-treated cells) 

We clarified this issue in Materials and methods section (lines 200-203). Moreover, we have corrected a mistake in the axis title (Relative expression 2^dCt to Relative expression 2-ΔΔCt). Same method was used to calculate the relative expression in transfected cells using miR-scrambled or anti-miR-scrambled as controls.

Figure 2b-c: The increase in resistance to vemurafenib after overexpression or inhibition of miR-410-3p is not seem in all the cell lines tested, this can not be ignored and should be mentioned in the text.

Authors’ response:

We have mentioned this issue in the text (lines 373-377).

Figure 2: Transient transfection efficiency should be shown: if the experiments are performed 48h after transfection, results showing changes in miR-410-3p expression, either its increase with the mimic or its inhibition with the anti-miRNA compared to the corresponding controls (miR-scr or anti-miR-scr, respectively), should be included at this time point.

Authors’ response:

We have added S2 Fig showing the transfection efficiency 48h after transfection. The results are presented as relative expression compared to the corresponding controls (miR-scr or anti-miR-scr, respectively) and calculated using 2-ΔΔCt method (lines 187-189).

Figure 3c: It would strengthen the results if the expression of some ER stress markers were analyzed by qPCR to prove that ER stress had been successfully achieved with the conditions used in relation to thapsigargin.

Authors’ response:

We have analyzed the expression of ER stress markers (CHOP, ATF4, and sXBP1) by qPCR. Their levels were upregulated in all experiments with ER stress. The results are presented in S4 Fig. Moreover, we have added results that confirm the induction of ER stress by vemurafenib (S3 Fig).

Figure 3c: it would be interesting to perform an experiment with one of the cell lines comparing the expression of miR-410-3p in response to vemurafenib, thapsigargin and the combination of both.

Authors’ response:

We thank Reviewer for the suggestion. We have performed additional experiments to test this issue. We have found that vemurafenib in combination with ER stress inductor (thapsigargin) more potently induces miR-410-3p expression. The results are presented in figure 3d.

Figure 4b: changes in AXL expression are only statistically significant after overexpressing miR-410-3p in the A375 cell line, this should be mentioned in the text.

Authors’ response:

We have mentioned this issue in the text (lines 291-292, 383, 385-386).

Figure 4b: the expression levels of AXL in the transfectant controls are missing (miR-scrambled and anti-miR-scrambled).

Authors’ response:

Figure 4b presents the relative expression of AXL, calculated as 2-ΔΔCt. The AXL levels were calculated as relative expression compared to the corresponding controls (miR-scrambled for mimic-miR-410-3p or anti-miR-scrambled for anti-miR-410-3p). We have clarified this in the figure legends, methods section and we have corrected the axis title.

Figure 5 (upper left): Most of miRNAs expression changes show here seem to be modified after BRAF inhibitor treatments. Apparently, by what is shown in the figure and how it is designed, no differences in most of miRNA expression occurs after treatment with MEK and ERK inhibitors. This is inaccurate, as for example, miR-410-3p expression in response to MEK and ERK inhibitors has not been analysed in this study, which raises the possibility that the same is happening in the others studies mentioned here. This information is relevant and it must be included in the scheme for all the miRNAs reported according to the studies used to create this figure.

Authors’ response:

We have modified Fig 5 and focused only on our results.

Figure 5 (bottom right): The upregulation of miR-410-3p expression can not be related to a downregulation of MITF expression as this issue has not been addressed anywhere in this study, remove it or perform the corresponding experiments to claim so.

Authors’ response:

We have remove it from the scheme and we hope that now it is more clear.

In general, the writing quality of the manuscript could be improved, and authors should revised it.

Authors’ response:

We have corrected all grammatical mistakes in the manuscript and we have done our best to improve the writing quality. 

Figure 1a: the meaning of the abbreviations for the types of cancer analysed are missing.

Authors’ response:

We have added the meaning of the abbreviations to the figure legend.

Figure 1b: the second highest enrichment pathway for miR-410-3p targets is the p53 signalling pathway. It would be of interest to include this finding in the discussion, along with any hypothesis in relation with the present study.

Authors’ response:

We have discussed this issue in the text (lines 323-324, 351-353).

Figure 1c-line 169: as the differences in survival (Figure 1c) does not seem to be significant (p=0.0764), the sentence addressing this result could be less strong (such as “there is a slight association”). Also, the period of time when this difference is higher could be specifically mention in the text.

Authors’ response:

We have modified this statement according to the Reviewer’s suggestion (lines 222-223).

Line 176: the BRAF mutation status of the cell lines used should be mentioned.

Authors’ response:

We have mentioned the BRAF mutation status of the cell lines (lines 228-229).

Figure 4a-methods: the statistical method used to analyze the correlation of AXL and miR-410-3p expression must be included in the methods section.

Authors’ response:

We have added the information about the analysis of the correlation of AXL and miR-410-3p expression in the methods section (lines 132-134).

Figure 4: the results about the switching towards a more invasive phenotype due to miR-410-3p expression would be strengthened if any marker of the proliferative phenotype would be characterised when overexpressing or inhibiting this miRNA.

Authors’ response:

We have analyzed the expression of MITF and we have observed a slight decrease, however, without statistical significance. We plan to asses the role of miR-410-3p in the regulation of invasiveness and proliferation of melanoma cells in the future.

Discussion: the low miR-410-3p expression found in the tumours sections of melanoma patients samples (Figure 1d) should be further discussed in relation to its significance in the context of the upregulation of this miRNA after vemurafenib treatment in human melanoma cell lines. Also, if the patients included in this study were treated or not should be clearly stated.

Authors’ response:

We have discussed this issue more deeply in revised manuscript. Also, we have added the information about patients to the methods section.

Reviewer #2

The manuscript by Paskal and coauthors reports about the involvement of miR410 in melanoma resistance to BRAF targeted drugs. The authors show that miR410 is induced by the drug in cell lines and its inhibition or ectopic expression associates with in vitro drug sensitivity. Analysis in a set of melanoma primary tumors showed that miR410 expression is lower in melanoma cells compared to adjacent skin tissue, and in silico analysis showed that miR410 expression correlates with markers associated to drug resistance.

The study is clearly written and reports interesting results, adding miR410 to previously reported miRs associated to melanoma resistance to the effect of BRAF targeted drugs. Nonetheless, I have few points that need clarification.

Authors’ response:

We thank Reviewer for appreciation our work and for the suggestions that improved our manuscript. We hope that we have clarified all issues raised by Reviewer.

- The authors should explain how they identified miR410 for their study

Authors’ response:

We have checked the expression of several miRNAs after vemurafenib treatment of melanoma cells. We focused on the miRNAs that were described as important in the regulation of tumor cells resistance to different types of the therapies. We found that miR-410-3p were strongly induced in melanoma cells after vemurafenib treatment. Therefore, we decided to focus on this microRNA.

- Please explain more about the importance of mir410 and its mechanism in the introduction section (according to previous studies)

Authors’ response:

We have discussed the importance of miR-410-3p in the introduction section of revised manuscript (lines 81-94).

- Table 1 reports clinical data about the studied tumors which are not of interest in the results section, and should be reported as supplementary information

Authors’ response:

We have moved Table 1 into supplementary data (S1 Table).

- Fig 5 should be better focused on miR410 effects and mechanism

Authors’ response:

We have modified Fig 5 and we have focused only on miR-410-3p.

- Supplementary table 1 lacks a title and an explanation, and why some lines are in bold

Authors’ response:

We have added the title and an explanation to the Supplementary Table 1 – S2 Table in revised manuscript.

- Fig 3a and 4a are poorly readable

Authors’ response:

We have modified the size of the panels in the Fig 3a and Fig 4a to improve their quality.

- The authors should include the control gene used for miR410 level analysis by qPCR (fig 1e).

Authors’ response:

We have added the information about control gene (U6) used for miR-410-3p level analysis by qPCR to the materials section.

---

## [Decision Letter · Decision Letter 1]

2 Jun 2020

miR-410-3p is induced by vemurafenib via ER stress and contributes to resistance to BRAF inhibitor in melanoma

PONE-D-20-03181R1

Dear Dr. Paskal,

We are pleased to inform you that your manuscript has been judged scientifically suitable for publication and will be formally accepted for publication once it complies with all outstanding technical requirements.

With kind regards,

Salvatore V Pizzo

Academic Editor

PLOS ONE

Additional Editor Comments (optional):

Reviewers' comments:

Reviewer's Responses to Questions

**Comments to the Author**

1. If the authors have adequately addressed your comments raised in a previous round of review and you feel that this manuscript is now acceptable for publication, you may indicate that here to bypass the “Comments to the Author” section, enter your conflict of interest statement in the “Confidential to Editor” section, and submit your "Accept" recommendation.

Reviewer #1: All comments have been addressed

Reviewer #2: All comments have been addressed

2. Is the manuscript technically sound, and do the data support the conclusions?

Reviewer #1: (No Response)

Reviewer #2: (No Response)

3. Has the statistical analysis been performed appropriately and rigorously? 

Reviewer #1: (No Response)

Reviewer #2: (No Response)

4. Have the authors made all data underlying the findings in their manuscript fully available?

Reviewer #1: (No Response)

Reviewer #2: (No Response)

5. Is the manuscript presented in an intelligible fashion and written in standard English?

Reviewer #1: (No Response)

Reviewer #2: (No Response)

6. Review Comments to the Author

Reviewer #1: The authors have addressed all the points raised, improving the quality of the manuscript and strengthening its results.

Reviewer #2: (No Response)

7. PLOS authors have the option to publish the peer review history of their article (what does this mean?). If published, this will include your full peer review and any attached files.

Reviewer #1: No

Reviewer #2: No

---

## [Editor Report · Acceptance letter]

8 Jun 2020

PONE-D-20-03181R1 

miR-410-3p is induced by vemurafenib via ER stress and contributes to resistance to BRAF inhibitor in melanoma 

Dear Dr. Paskal:

I'm pleased to inform you that your manuscript has been deemed suitable for publication in PLOS ONE. Congratulations! Your manuscript is now with our production department. 

Kind regards, 

on behalf of

Dr. Salvatore V Pizzo 

Academic Editor

PLOS ONE